# Transcriptomic Analysis of Testicular Gene Expression in a Dog Model of Experimentally Induced Cryptorchidism

**DOI:** 10.3390/cells11162476

**Published:** 2022-08-10

**Authors:** Hyunjhung Jhun, Won-Young Lee, Jin-Ki Park, Sun-Goo Hwang, Hyun-Jung Park

**Affiliations:** 1Food Industry Infrastructure Team, Korea Food Research Institute, Wanju-gun 55365, Korea; 2Department of Livestock, Korea National University of Agricultures and Fisheries, Jeonju-si 54874, Korea; 3Department of Smart Life Science, College of Life and Environment Science, Sangji University, Wonju-si 26339, Korea; 4Department of Animal Biotechnology, College of Life and Environment Science, Sangji University, Wonju-si 26339, Korea

**Keywords:** cryptorchidism, tight junction, claudin, sertoli cell, dog

## Abstract

Cryptorchidism, a condition in which testes fail to descend from the abdomen into the scrotum, is a risk factor for infertility and germ cell cancer. Normally, tight junctions between adjacent Sertoli cells in the testes form a blood–testes barrier that regulates spermatogenesis; however, the effect of cryptorchidism on tight junctions is not well-understood. We established a model of heat-induced testicular damage in dogs using surgical cryptorchidism. We sequenced RNA to investigate whether certain transcripts are expressed at higher rates in heat-damaged versus normally descended testes. Claudins, cell adhesion molecules, were relatively highly expressed in cryptorchid testes: claudins 2, 3, 5, 11, and 18 were significantly increased in cryptorchid testes and reduced by orchiopexy. *SOX9*-positive Sertoli cells were present in the seminiferous tubules in both cryptorchid and control testes. Using real-time PCR and Western blot analysis to compare Sertoli cells cultured at 34 °C and 37 °C, we found that Sertoli cell claudins 2, 3, 5, 11, and 18 were significantly increased at 37 °C; however, accumulation was higher in the G0/G1 phase in Sertoli cells cultured at 34 °C. These results indicate that testicular hyperthermia caused by cryptorchidism affects claudin expression, regulated germ cell death, and the proliferation of Sertoli cells.

## 1. Introduction

In mammalian testes, the blood–testes barrier (BTB) is formed between adjacent Sertoli cells during puberty by a complex of different junction types. The BTB divides the seminiferous epithelium into a basal compartment containing spermatogonia and a luminal compartment containing more developed male germ cells, including spermatocytes, spermatids, and mature sperm [1]. Tight junctions (TJs) between Sertoli cells restrict the movement of water, solutes, and immune cells into seminiferous tubules, thus creating an immunologically unique microenvironment for spermatogenesis [2]. The BTB comprises TJs and TJ-associated proteins, such as claudins and occludins, as well as cadherin-based adherens junctions and connection-based gap junctions, which are structurally and functionally in contact with each other [3].

Claudin mRNAs that have been identified in testes via Northern blot analysis include claudins 1, 2, 3, 4, 5, 7, 8, and 11 [4,5,6]. In addition, claudins 10, 12, and 23 have been detected by microarray analyses in rodents [7]. Claudin-1 mRNA and protein are present in mouse testes at postnatal day 3, increased at postnatal day 10, but decreased on postnatal days 16 and 30, compared to levels at day postnatal 3 [8]. In rats, the expression of claudin-1 protein adjusted for testes weight increases from postnatal day 16–30 and then plateaus in adulthood [9]. Claudin-3 is first detected at postnatal day 15 in mouse testes and is localized to the BTB at postnatal day 20. In adult mouse testes, claudin-3 is specifically expressed at the seminiferous stage of spermatogenesis, in which preleptotene and leptotene spermatocytes migrate across the BTB [10]. Claudin-5 is expressed throughout the entirety of the seminiferous epithelium surrounding germ cells in postnatal day 8 mouse testes. In adult mouse testes, claudin-5 exhibits stage-specific expression in the BTB, with increased expression observed at stages VIII and IX of the spermatogenic cycle, similar to claudin-3 [10]. Claudin-11 mRNA expression is first detected prenatally in mouse testes at postcoital day 12 at approximately the same time as the formation of the seminiferous cords. Furthermore, in mice, when developmental claudin-11 mRNA expression is normalized to the number of Sertoli cells, the expression increases from postnatal day 0 through adulthood [11].

Spermatogenesis is a unique process in which spermatogonia, which are diploid cells, give rise to functional haploid sperm. The maintenance of an appropriate scrotal temperature is essential for normal spermatogenesis; for this reason, testes are carried below the abdomen in most male mammals [12]. Cryptorchidism, a condition in which one or both testes fail to descend from the abdomen into the scrotum, is the most common disorder of sexual differentiation as well as the most common disease of the male endocrine organs [13]. The term “cryptorchid” means hidden testes: when no testes are present in the scrotum, it is called bilateral cryptorchid, and when only one testes is present in the scrotum, it is known as unilateral cryptorchid. Unilateral cryptorchidism occurs more commonly than the bilateral condition [14].

In dogs with cryptorchidism, the reduction in testes size, low serum levels of testosterone [15,16], and lower expression levels of *CYP19A1*, which is a steroidogenesis-related gene, were observed in the undescended testes [17]. Cryptorchid testes have significantly higher temperature than normally descended testes [18]. Testes should be 4–5 °C cooler than the body temperature to produce normal sperm; therefore, unilateral cryptorchids reduce semen quality due to the effects of high body temperature on the retained testes [19]. The major consequence of heat stress on testes is known to be the destruction of germ cells through the reactive oxygen species (ROS)-mediated apoptotic signaling pathway with deleterious effects on sperm, spermatocytes, and spermatids [20].

Although several articles have discussed the correlation between cryptorchid-induced heat damage and BTB molecules, the specific mechanisms underlying this correlation are not well-understood. In this study, we established a model of surgically induced cryptorchidism in dogs, which is induced by heat damage, to investigate RNA expression patterns in cryptorchid compared to those in normal testes. Additionally, we aimed to confirm whether the changes occurring in the cryptorchidism models were recovered in orchiopexies models.

## 2. Materials and Methods

### 2.1. Study Animals and Surgical Procedures

Six 12-month-old male beagle dogs were used in this study. Cryptorchidism was surgically induced as previously described [21] with some modifications. The experimental design is outlined in Figure 1. Briefly, dogs received acepromazine (0.1 mg/kg; intramuscular injection (IM)) for premedication and propofol (6 mg/kg; intravenous injection) for induction of inhalation anesthesia. Endotracheal intubation was performed using a tube with an internal diameter of 7 mm, and anesthesia was maintained via inhalation of 2% isoflurane delivered in 100% oxygen. Each animal was administered cefazolin sodium (22 mg/kg; IM) at the time of induction of anesthesia. All these chemical reagents were purchases from Sigma Aldrich (St. Louis, MO, USA). Physiologic monitoring conducted during surgery and anesthesia included continuous pulse oximetry, electrocardiography, and continuous measurements of blood pressure variables, heart and respiratory rates, and body temperature via an esophageal probe. Meloxicam (0.1 mg/kg; subcutaneous injection) was used as a pre- and postoperative analgesic. All surgery and administration of anesthesia was performed by an experienced veterinarian.

On experiment day 1, we performed surgery to induce unilateral cryptorchidism, where one testicle did not descend in all dogs in the experiment. The testicle on which surgery was not performed was used as the wild-type control.

A 4 cm long paramedian incision was made through the skin in the inguinal region. Subsequently, the superficial and internal inguinal rings were exposed and enlarged by incision. The gubernaculum on the right side was then cut to displace the right testes into the abdomen, after which the inguinal canal was closed. The contralateral (left) testes was sham-operated to act as a paired control. Next, on experiment day 30, we performed laparoscopic sampling of wild-type and surgical-induced cryptorchid testes for RNA sequencing, RNA and protein analysis on three dogs. In addition, on experiment day 30, we performed orchiopexies on three dogs, in which the cryptorchid testes were returned to the scrotum through an inguinal incision and sutured to fix them there.

Finally, on experiment day 60, all testes were removed and decapsulated. The testes were stored at −80 °C for RNA and protein analyses and stored in 4% formaldehyde solution for histological and immunohistochemical analyses.

### 2.2. Isolation and In Vitro Culture of Sertoli Cells

We isolated Sertoli cells from the collected experimental and control testes based on the methods described in our previous study [22]. Briefly, the testes were decapsulated and mined using fine scissors. Five volumes (*v*/*w*) of enzyme A (0.5 mg/mL collagenase, 0.01 mg/mL DNase I, 0.1 mg/mL soybean trypsin inhibitor, and 0.1 mg/mL hyaluronidase) were added to testes samples, after which the samples were incubated at 37 °C for 10 min. After incubation, the samples were washed with phosphate-buffered saline (PBS), and then five volumes (*v*/*w*, original testes weights) of enzyme B (5 mg/mL collagenase, 0.01 mg/mL DNase I, and 0.1 mg/mL soybean trypsin inhibitor) were added, before being incubated at 37 °C a second time for 10 min. Samples were meshed using a 40 μm nylon mesh after washing with PBS. Red blood cells were eliminated using a lysis buffer (Sigma-Aldrich, St. Louis, MO, USA).

Sertoli cells were isolated by density gradient ultracentrifugation using a gradient of 20% and 40% Percoll (Sigma-Aldrich). A 40% Percoll suspension was prepared by mixing 40% Percoll solution, 10% PBS (*v*/*w*), 1% fetal bovine serum (FBS, *v*/*w*), 0.5% antibiotics (*v*/*w*, 50 U/mL, and 50 μg/mL penicillin and streptomycin, respectively), and 48.5% ultrapure water (*v*/*w*) in a 15 mL conical tube. A 20% suspension was prepared by diluting the iso-osmotic 40% Percoll suspension with PBS supplemented with 1% FBS (*v*/*w*) and antibiotics. The cell suspensions were then loaded on top of the gradient and centrifuged at 600× *g* for 15 min at 4 °C. The cells at the top layer (i.e., the 20% Percoll layer) were collected and used as Sertoli cells.

The isolated Sertoli cells were cultured at 34 and 37 °C in Dulbecco’s modified Eagle’s medium supplemented with 10% FBS and 1% penicillin/streptomycin. Confluent cells were washed with PBS and then detached from culture dish by Trypsin–EDTA solution. PBS was added to detached cells, and they were subjected to centrifugation at 200× *g* for 5 min. The supernatant was then discarded, and the cells were resuspended in culture medium.

### 2.3. Hematoxylin and Eosin Staining

Fixed samples were washed with 70–100% (*v*/*v*) ethanol, embedded in paraffin, sliced into 5 µm thick sections using a microtome (Thermo Fisher Scientific, Inc., Waltham, MA, USA), and mounted on glass slides. The mounted tissues were subsequently rehydrated with xylene and 100% ethanol and stained with hematoxylin and eosin (Sigma-Aldrich). Representative images of stained testes sections were obtained using a Nikon E-800 microscope (Nikon E-800, Tokyo, Japan) and Motic Images Advanced 3.2 software (Kowloon, Hong Kong).

### 2.4. Immunohistochemistry and Immunocytochemistry

Tissue sections were rehydrated using xylene and 100–50% ethanol. Antigen retrieval was performed by boiling the samples in 10 mM citrate buffer (pH 6.0) for 10 min. Nonspecific binding was blocked using 2% bovine serum albumin (BSA) and 0.05% Triton X-100 in PBS for 30 min at 20 °C. The samples were incubated overnight at 4 °C with the primary antibodies. After washing three times with PBS, the appropriate secondary antibodies were added and left to incubate for 1 h at room temperature. To identify nuclei, 4′,6-diamidino-2-phenylindol (DAPI; Sigma–Aldrich) was then added at a concentration of 1 µg/mL and left to react for 10 min. Finally, a mounting solution (Dako Fluorescent Mounting Medium; Dako North America, Inc., Carpinteria, CA, USA) was added. For SOX9 antibody staining, samples were incubated with SOX9 antibodies at 4 °C for 18 h, after which they were washed with PBS, then incubated for 1 h at room temperature with secondary antibodies. A peroxidase substrate detection kit (SK-4100; Vector Laboratories, Burlingame, CA, USA) was used.

For immunocytochemistry, the Sertoli cells that were isolated in Section 2.2 were cultured in vitro. These cells were washed three times with PBS and fixed in 4% paraformaldehyde for 10 min. Membrane permeabilization was then performed for 10 min with PBS containing 0% Triton X-100. Nonspecific protein binding was blocked with 2% BSA in PBS for 30 min at room temperature. The cells were incubated overnight at 4 °C with the primary antibodies, washed with PBS, and subsequently incubated for 1 h at room temperature with secondary antibodies. A peroxidase substrate detection kit (SK-4100; Vector Laboratories, Burlingame, CA, USA) was used to detect vimentin according to the manufacturer’s instructions. The antibodies used to analyze the expression of claudins and other marker proteins are listed in Table 1. Images were taken using a Nikon E-800 microscope.

### 2.5. RNA Extraction and Real-Time PCR

RNA was extracted from three different regions in normal, cryptorchid, and orchiopexied dog testes using the Qiagen RNA extraction kit, according to the manufacturer’s instructions (74104; Qiagen, Venlo, The Netherlands). Genomic DNA was removed with DNase I treatment according to the manufacturer’s instructions (Qiagen). We then synthesized cDNA from 1 μg of total RNA using an RT-PCR premix kit (15131; iNtRON, Seongnam, Korea). The relative amounts of claudin mRNAs were estimated in duplicate samples by fluorescence and quantified using the Qiagen Rotor-Gene PCR detection system. The reaction was initiated in a total volume of 20 μL, which contained 10 ng of cDNA and 1 pM of each primer set in a reaction buffer containing iQ SYBR Green Supermix (170–8880; Bio-Rad Laboratories, Hercules, CA, USA). Cycle threshold (Ct) values were normalized to glyceraldehyde 3-phosphate dehydrogenase (*GAPDH*) gene expression. The real-time PCR results were expressed as target gene expression relative to the expression of the control gene (*GAPDH*). PCR amplification was performed using 40 cycles of 20 s at 95 °C, 20 s at 55 °C, and 20 s at 72 °C. The primers used for the expression of claudins and other marker genes are listed in Table 2.

### 2.6. Western Blotting

Total protein from the testes samples and in vitro cultured Sertoli cells was isolated using a Proprep kit (17081; iNtRON) according to the manufacturer’s instructions. Fifty micrograms of total protein from each sample were separated using 4–20% gradient SDS-polyacrylamide gel electrophoresis (Bio-Rad Laboratories) and transferred to polyvinylidene fluoride membranes. The membranes were blocked with 5% non-fat milk and incubated for 1 h at room temperature with the primary antibodies for each protein: β-actin and clusterins 2, 3, 5, 11, 18. After washing three times with tris-buffered saline with Tween™ 20 detergent (Thermo Fisher Scientific, Inc.), after which the appropriate secondary antibodies were added. The membranes were then incubated for 2 h at room temperature, after which protein expression was confirmed using enhanced chemiluminescence (32106; Thermo Fisher Scientific, Inc.) with HyBlot CL^®^ Autoradiography Film (No. E3018; Denville Scientific, Metuchen, NJ, USA). The antibodies used for Western blot analysis of claudins and other marker proteins are listed in Table 1.

### 2.7. Gene set Enrichment Analysis (GSEA)

To compare genes with predefined gene sets, genome wide annotation for canine was downloaded from Bioconductor (https://bioconductor.org/packages/release/data/annotation/html/org.Cf.eg.db.html. accessed on 5 September 2020). The gene set enrichment analysis (GSEA) was performed using R package clusterProfiler (https://bioconductor.org/packages/release/bioc/html/clusterProfiler.html. accessed on 8 September 2020) with gseGO, enrichGO, and gseKEGG functions, and then visualized using R package ggplot2 (https://cran.r-project.org/web/packages/ggplot2/index.html. accessed on 8 September 2020), enrichplot (https://bioconductor.org/packages/release/bioc/html/enrichplot.html. accessed on 9 October 2020), and pathview (https://bioconductor.org/packages/release/bioc/html/pathview.html. accessed on 9 October 2020).

### 2.8. Statistical Analysis

The gene and protein expression data were analyzed by one-way analysis of variance (ANOVA), using IBM^®^ SPSS^®^ Statistics ver. 21.0 for Windows (IBM, Armonk, NY, USA). Significant differences between two groups were determined using Student’s *t*-test*,* whereas differences among three or more groups or time points were determined by one-way ANOVA, followed by Tukey’s honestly significant difference test. All data are expressed as mean ± standard deviation. Values of * *p* < 0.05 were considered statistically significant, and for multiple comparisons, statistical differences among the groups are indicated using ‘a, b, and c’.

## 3. Results

### 3.1. Analysis of Germ Cells in Wild-Type, Cryptorchid, and Orchiopexied Dog Testes

To investigate germ cell depletion in cryptorchid and orchiopexied dog testes, we analyzed morphological changes and performed gene expression analyses. Most testicular cells in the seminiferous tubules were depleted in cryptorchid testes, although a few DAPI-stained cells were observed near the basement membrane of the seminiferous tubules. Testicular cells were relatively more abundant in orchiopexied testes than in cryptorchid testes; however, testicular cells were most abundant in wild-type testes relative to the other two groups (Figure 2A).

We conducted an immunostaining analyses to assess the expression of synaptonemal complex protein 3 (SYCP3), retinoic acid 8 (STRA8), and VASA, all of which had meiotic markers that are expressed in spermatocytes in testes tissues. SYCP3-positive spermatocytes were abundant in wild-type testes but not in cryptorchid testes. These results indicate that few cells within the tubules in cryptorchid testes are not spermatocytes. SYCP3-positive cells were also found in orchiopexied testes (Figure 2B). The expressions of SYCP3, STRA8, and VASA were significantly and consistently decreased in cryptorchid testes compared to wild-type and orchiopexied testes (Figure 2C). These results indicate that male germ cells were significantly reduced by surgically induced cryptorchidism and that testes function and spermatogenesis resumed after orchiopexy.

### 3.2. Analysis of Sertoli Cells in Wild-Type, Cryptorchid, and Orchiopexied Dog Testes

Immunostaining analysis of SOX9 and VASA as a marker of Sertoli cells and germ cells in dog testes were performed (Figure 3A,B). SOX9-positive Sertoli cells were clearly observed in all wild-type, cryptorchid, and orchiopexied dog testes (Figure 3A). However, VASA-positive germ cells were observed in wild-type and orchiopexied testes but not in cryptorchid testes (Figure 3B). Additionally, the number of observed SOX9-positive cells did not differ between the three groups (Figure 3C). In addition, the expression levels of the Sertoli cell marker genes *SOX9*, *WT1*, and *AMH* were analyzed using real-time PCR. The expressions of *SOX9*, *WT1*, and *AMH* did not significantly differ between cryptorchid, orchiopexied, and wild-type testes (Figure 3D–F).

### 3.3. RNA Sequencing and Gene Enrichment Analysis

To assess the functional alteration of genes after treatment, we detected the differentially expressed genes (DEGs) using RNA-seq analysis (Appendix A). Over 91% of reads in each sample were mapped to a reference genome for *Canis lupus familiaris*. The percentage of multiple mapped reads in total reads was low and ranged from 3.1% to 7.8%. Moreover, we observed high correlations of genes in three replications for WT or CT, indicating the proper utilization of RNA-seq data in this study (Appendix A). A total of 4448 DEGs showed 2102 upregulations and 2346 downregulations in CT compared to WT (Appendix A).

To understand the altered gene ontology (GO) biological processes, including the activated or suppressed function between WT and CT, we performed a GSEA (Figure 4A). The activated biological processes were related to responses to cytokine stimulus, steroid esterification, lipid and sterol transport, digestion, cell–cell adhesion, etc. Meanwhile, reproduction-related GO functions were suppressed in CT. In particular, biological processes such as spermatogenesis, male gamete generation, and germ cell developments showed a high statistical significance with lower adjusted *p*-values. We confirmed the gene set enrichment at the front of sequence (over 0.8 enrichment score) in cell–cell adhesion via plasma-membrane adhesion molecules; however, spermatogenesis had an enrichment score less than −0.8 at the back of sequence (Figure 4B). Various connections and regulation mechanisms among the genes involved in the most prominent GO biological process were observed (Figure 4C). The DEGs were divided into two groups of GO functions pertaining to reproduction and cell projection. Several genes involved in the five prominent GO functions (multi-organism reproductive process, sexual reproduction, male gamete generation, cell projection assembly, and plasma membrane-bounded cell projection assembly) were down-regulated in CT. Seven genes (*RSPH1*, *TTC26*, *MTOR*, *SPATA6*, *NEURL1*, *TEKT3*, and *TSGA10*), which act as bridges linking the two prominent GO functions, were also downregulated in cryptorchid testes compared to that in wild-type testes. In the connection between DEGs, the *CLDN11* gene encoding the cell-adhesion molecule claudin-11 showed an upregulation in CT, suggesting that the *CLDN* gene may play a negative role in reproduction-related pathways such as spermatogenesis and male gamete generation.

To determine whether tight junction- and necroptosis-related genes were affected in cryptorchid testes, we confirmed the altered gene expressions of DEGs using KEGG pathway analysis (Figure 5). In the tight junction pathway, the upregulation of DEGs in CT seemed to induce cell polarity, decreased paracellular permeability, and tight junction and actin assembly. Several genes involved in the necroptosis pathway were upregulated, and this seemed to induce the dissipation and the transient increase in the mitochondrial transmembrane potential and increased lysosome membrane permeabilization.

### 3.4. Quantification and Analysis of Clusterins in Wild-Type, Cryptorchid, and Orchiopexied Dog Testes

To verify the RNA sequencing data discussed in Section 2.3 pertaining to dog testes samples, the expression levels of claudins 2, 3, 5, 11, and 18 were assessed in three different groups using real-time PCR and Western blot analyses. The expression of claudins 2 and 5 significantly increased in cryptorchid testes relative to orchiopexied and wild-type testes; however, claudins 2 and 5 were relatively decreased in orchiopexied testes compared to the wild-type (Figure 6A,C). The expressions of claudins 3 and 18 relatively increased in both cryptorchid and orchiopexied testes, but these experessions were much higher in cryptorchid than orchiopexied groups (Figure 6B,E). The expression of claudin-11 was relatively increased in cryptorchid testes and returned to the orchiopexied group (Figure 6D). Consistently, the relative expression of claudins 2, 3, 5, 11, and 18 was distinctly higher in cryptorchid testes than in wild-type testes, returned in orchiopexied testes, although the rate of decrease was different for each claudin (Figure 6F).

In addition, the relative expression of claudins 3 and 18 in wild-type, cryptorchid, and orchiopexied dog testes was assessed by immunostaining using commercially available antibodies. Claudins 3 and 18 were present in the cell membrane near the basement membrane. Furthermore, cells that were positive for claudins 3 and 18 were also partially colocalized with Sertoli cells that were marked with vimentin (Figure 7).

### 3.5. Expression of Claudins in In Vitro Cultured Dog Primary Sertoli Cells

To verify claudin expression at different temperatures, primary dog Sertoli cells were isolated and cultured at 34 °C and 37 °C. Expression of vimentin, a marker of Sertoli cells in testes and in vitro cultured Sertoli cells, was confirmed by immunocytochemistry analysis (Figure 8A). The morphology of in vitro cultured Sertoli cells was changed by different temperature conditions after 4 days in an in vitro culture. The Sertoli cells grew well at 37 °C, and morphological changes were detected. The morphology of cells grown at 37 °C was longer than that of cells grown at 34 °C (Figure 8B). Next, the relative expression of different claudins was compared in cells grown at 37 °C and 34 °C after a 7-day culture period. Relative expression of claudins 2, 3, 5, 11, and 18 was quantified by real-time PCR and Western blot analysis. Expression levels of *Claudin−2*, −*3*, −*5*, −*11*, and −*18* genes were significantly increased in cells grown at 37 °C compared to those grown at 34 °C. The protein analysis results were similar to the real-time PCR results (Figure 8C). Consistently, the protein levels of claudins 2, 3, 5, and 11 were also highly detected in cells grown at 37 °C compared to those grown at 34 °C (Figure 8D).

## 4. Discussion

In most mammals, testicular temperature is consistently lower than the core body temperature in order to maintain normal spermatogenesis. Exposure of the testes to body temperature due to local testicular hyperthermia, cryptorchidism, and varicocele results in increased germ cell death [23]. However, molecular mechanisms underlying testes responses to heat stress and the association with infertility in dogs have not yet been fully elucidated. Here, we focused on the relative expression patterns of claudins in dog testes with experimentally induced cryptorchidism to elucidate their correlation with male infertility caused by heat stress.

TJs between adjacent Sertoli cells are responsible for establishing the BTB, which divides the seminiferous epithelium into luminal and basal compartments. TJs are multi-molecular membrane specializations comprising integral membrane proteins such as claudins, occludin, and peripheral proteins, including ZO-1, 2, and 3 [24,25]. Our previous study showed that Sertoli cells are not damaged by cryptorchidism in dogs [21]. Based on these results, differentially expressed genes in wild-type and cryptorchid dog testes were identified using total RNA sequencing analysis. We found that claudins were highly expressed in cryptorchid dog testes. The potential use of claudins as markers of male infertility and their putative functions are discussed in this section.

Interestingly, our RNA sequencing data show that claudins 2, 3, 5, 10, 11, 12, 18, and 20 are highly expressed in surgically induced cryptorchid testes compared to that in wild-type testes, although claudins 1, 8, 10, 11, 12, and 23 are found in rodent testes [4,5,6]. Hellani et al. reported that the expression levels of claudin-11 in the testes increased during pre-pubertal development but gradually decreased according to testicular maturation because of the increased expression of germ cells that do not express claudin-11 [26]. Similarly, our results also show that the expression of claudin-11 was increased in cryptorchid dog testes, which lack endogenous germ cells. Testosterone is important for maintaining BTB integrity [27]. The effects of testosterone and androgen receptor (AR) signaling on mRNA expression, protein expression, and protein localization of TJ proteins, including claudins 1, 3, and 11, have been the focus of previous studies [24]. In primary cultures of mouse Sertoli cells, testosterone supplementation and co-culture with Leydig cells do not significantly affect claudin-1 expression [8]. In addition, claudin-3 dependence on AR signaling was detected in Sertoli cell-specific AR knockout mice, in which claudin-3 was reduced in testes of 8 weeks old mice [10]. In contrast, our results show an increase in claudin-3 expression in cryptorchid dog testes, which have lower testosterone levels than the wild-type [16].

Anti-Müllerian hormone (AM), a glycoprotein secreted from the fetal testes, is responsible for the regression of the Müllerian duct in the male fetus. In humans, progressive maturation of Sertoli cells, as shown by the decrease in AMH expression, with increasing age and AMH expression, was negatively correlated with the expression of connexion 43 and claudin-11 [28]. AMH and serum AMH concentrations may be useful biomarkers for the diagnosis of cryptorchidism [29]. In the present study, claudin-11 was highly expressed in cryptorchid testes, but AMH expression levels were not significantly changed by cryptorchidism. This result indicates that the expression of claudins is not correlated with AMH expression, which is a putative marker of cryptorchidism in dogs different from humans. There is evidence that the levels of serum AMH do not change between cryptorchid and wild-type dog testes. Gharagozlou et al. measured the levels of AMH in cryptorchid, castrated, and wild-type dogs and performed comparative analyses. AMH concentration was lower in castrated dogs than in the wild-type, but the concentration of AMH did not differ between wild-type and cryptorchid dogs [30].

Several reports described the correlation between claudins and oncogenesis. Claudin-1 is oncogenic in gastric cancer, and its malignant potential may be attributed in part to the regulation of anoikis by mediating membrane β-catenin-regulated cell-to-cell adhesion and cell survival [31]. The downregulation of claudin-3 is associated with proliferative potential in early gastric cancer [32]. Membranous claudin-4 expression is associated with gastric cancer progression and prognosis in gastric carcinoma [33]. In addition, increased expressions of claudin 6, 7, or 9 are able to enhance the migration, invasion, and proliferation of gastric cancer cells [34]. Furthermore, claudin-11 inhibits cell migration and invasion in nasopharyngeal carcinoma, and hypermethylated claudin 11 is associated with metastasis of colorectal cancer [35,36]. Among men with undescended testes, the risk of cancer is increased 2–8 times, and 5–10% of all men with testicular cancer have a history of cryptorchidism [37]. However, it is unknown whether cryptorchidism and testicular cancer share a common cause or whether cryptorchidism is itself a cause of testicular cancer. The change in claudin expression in cryptorchid testes may be associated with testicular cancer induction, as cryptorchidism is known as a risk factor for infertility and testicular germ cell tumors (TGCT) [38].

Yang et al. reported heat-treatment-induced oxidative damage in immature pig Sertoli cells via changes in tight junctions compared to in vitro cultured primary cells [39]. Our results show that the expression levels of claudins were higher in cultured Sertoli cells at 37 °C than in those cultured at 34 °C. However, contrary to our results, oxidative stress was reported to participate in the heat-stress-induced (42 °C) downregulation of tight junction proteins in cultured pig Sertoli cells via the inhibition of the CaMKKβ-AMPK axis; however, the mutation condition of Sertoli cells and culture temperature in this study differed from that employed in our experiments [40]. Another study showed spermatogenic abnormality in mouse testes exposed to a hot water at two distinct temperatures (39 or 42 °C) for 20 min, which is very short time period. While a temperature of 39 °C had no apparent influence on germ cell morphology and testicular histology, 42 °C did perturb spermatogenesis and spermatogenic subtypes [41]. Compared to that noted in our experimental results, it can be seen that the effects of heat damage on testes are sensitive to the time and temperature of heat exposure.

As mentioned above, a cryptorchidism model is associated with oxygen stress, as reactive oxygen species (ROS) are generated during heat stress and cause spermatogenic cell death [42]. The relationship between oxygen stress and claudin expression has been described in several studies. Although not testicular cells, Hirata et al. reported that the expression of TJ claudin-1, but not ZO-1, is increased in cells under H_2_O_2_-induced oxidative stress [43]. Another study also reported an upregulation of claudin-1 expression in hepatoma cells by ROS [44]. Interestingly, in rodents, local radiation-induced testicular oxidative stress increases the transcription levels of claudin-11 in the testes [45]. Additionally, Cai et al. reported that TJ occludin, claudin-3, and ZO-1 were reduced in testes after scrotal heat exposure (43 °C for 30 min); conversely, expression levels of claudin-11 were increased in the testes [44]. Although there are species differences in claudin expression in rodents and dogs, heat stress in cryptorchid testes leads to oxidative stress, which may be related to changes in claudin expression in the testes.

In conclusion, we performed a comparative study of surgically induced cryptorchid, orchiopexied, and wild-type testes in dogs, although the surgically induced cryptorchid model was not identical to spontaneous cryptorchidism. In particular, changes in gene expression in the testes caused by heat damage were studied using a surgically induced cryptorchid dog model; expression levels of claudins and germ cell death increased but gene expression in Sertoli cells remained unchanged. The findings of this study, for the first time, describe a possible correlation between claudin expression and heat stress in surgically induced cryptorchid testes in dogs. These results suggest that claudin, a type of tight junction protein, is closely related to the disruption of normal spermatogenesis due to heat damage. We can predict that the protein expression of claudin in Sertoli cells increased, thereby interfering with the differentiation and migration of undifferentiated germ cells for spermatogenesis; however, the detail mechanisms underlying claudin functions are not yet fully understood and further studies are needed. In addition, our study also indicates that claudin might be considered as a potential biomarker of detection of heat stress damage that can induce infertility.

## Figures and Tables

**Figure 1 cells-11-02476-f001:**
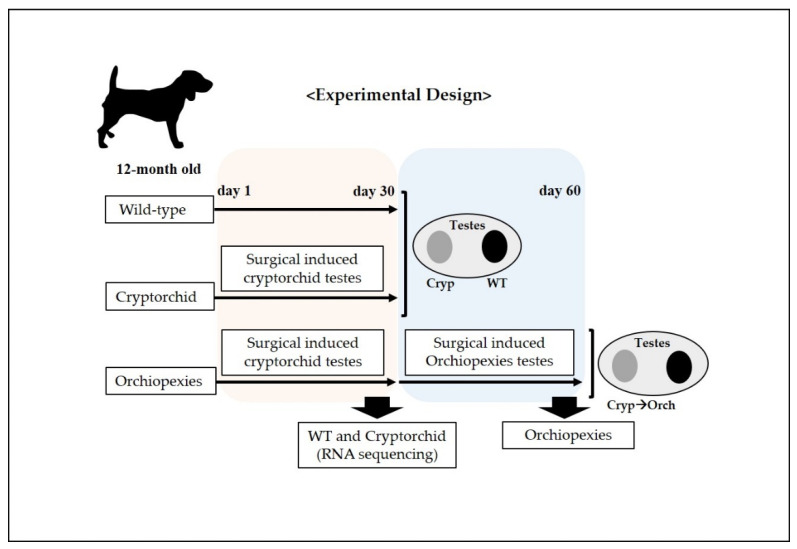
Schematic representation of the experimental design for the surgically induced cryptorchid and orchiopexies dog models. At day 1, cryptorchidism was surgically induced in one testicle in each dog (6 dogs), and the unoperated testicle was considered to be the wild type; cryptorchid samples were collected. In addition, 30 days after the surgical induction of cryptorchidism, surgery was performed to induce orchiopexies in 3 dogs; 30 days post-surgical induction of orchiopexies, testes samples were harvested.

**Figure 2 cells-11-02476-f002:**
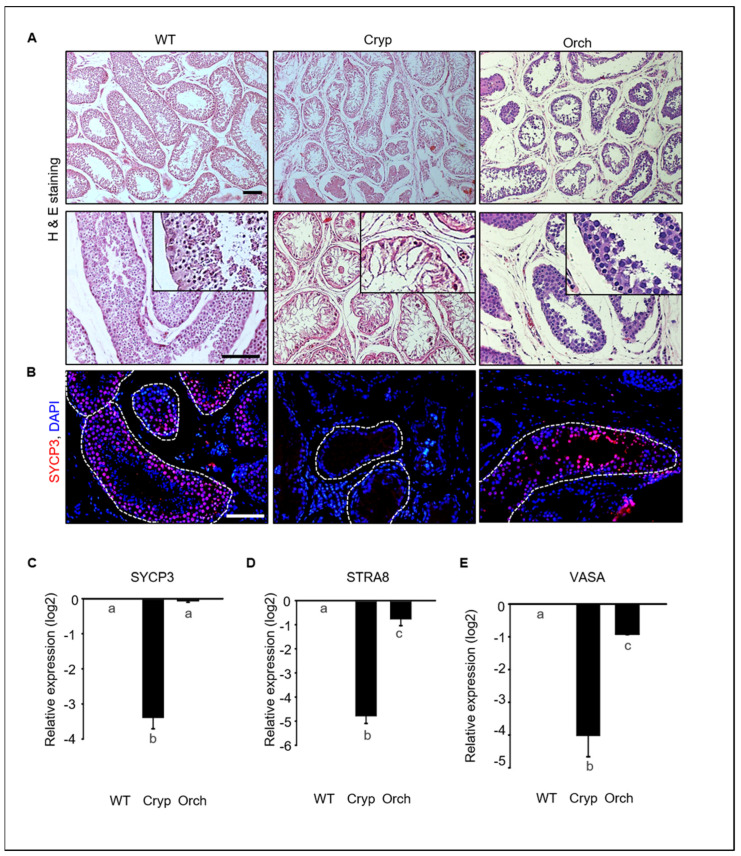
Germ cell depletion in cryptorchid dog testes. (**A**) Morphological changes in wild-type, cryptorchid, and orchiopexy dog testes analyzed using hematoxylin and eosin staining. Many germ cells were depleted in cryptorchid testes compared to wild-type and orchiopexied testes. (**B**) SYCP3, a meiotic marker protein, and DAPI were detected in wild-type, cryptorchid, and orchiopexied canine testes by immunostaining. SYCP3-positive cells were found in both wild-type and orchiopexied dog testes but not in cryptorchid testes. Scale bars = 50 μm; *n* = 3, two pairs of testes. (**C**–**E**) Expressions of *SYCP3*, *STRA8*, *VASA*, and *GAPDH* genes in wild-type, cryptorchid, and orchiopexied dog testes were determined by qPCR. The expression levels of *SYCP3*, *STRA8*, and *VASA* genes were decreased in cryptorchid testes compared to the wild-type. Data are presented as means ± standard deviation, and values with different letters are significantly different.

**Figure 3 cells-11-02476-f003:**
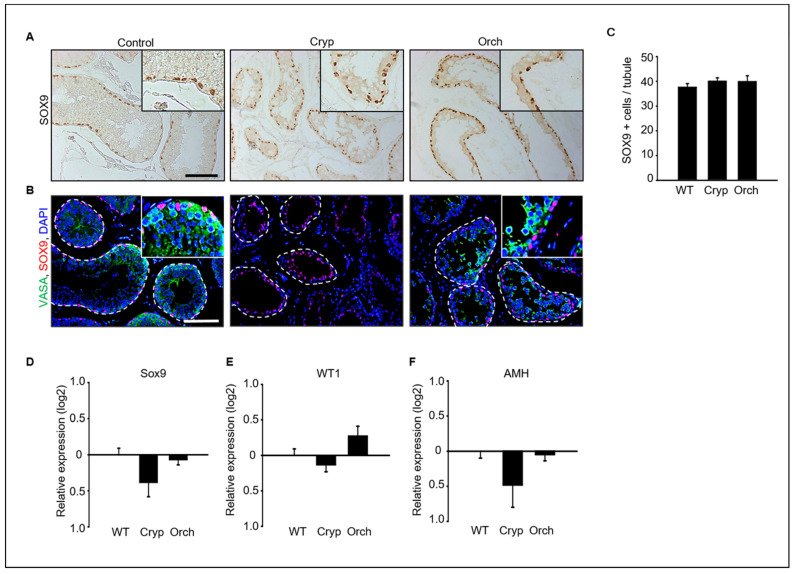
(**A**) SOX9, a Sertoli cell marker protein, was detected in wild-type, cryptorchid, and orchiopexied dog testes by DAB staining. SOX9-positive cells were found in all samples. (**B**) Both SOX9 and VASA, a germ cell marker, were detected in all three groups. VASA- and SOX9-positive cells were not exactly overlapped in wild-type testes; VASA-positive cells were not observed in cryptorchid testes. Scale bars = 50 μm; *n* = 3, two pairs of testes. (**C**) The average number of SOX9-positive Sertoli cells in the tubules was counted based on immunostaining from three different samples. At least 40 tubules were scored for each slide, for a total of 3 biological replicates. Data are presented as the mean ± standard deviation; no significant difference was found. Status of Sertoli cells in wild-type, cryptorchid, and orchiopexied dog testes. The expression levels of (**D**) *SOX9*, (**E**) *Wt1*, and (**F**) *AMH* genes were detected by qPCR.

**Figure 4 cells-11-02476-f004:**
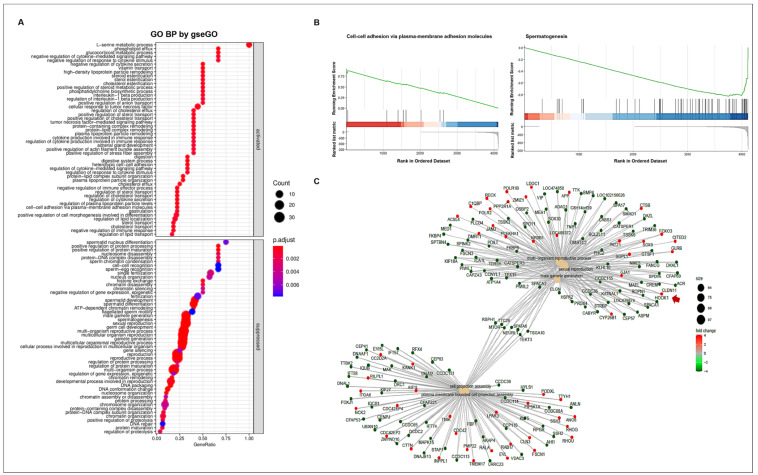
GO enrichment analysis. (**A**) The enriched GO biological processes. The GO annotations were divided into two groups namely activated and suppressed functions by R package clusterProfiler with gseGO function. The *x*-axis represents the gene ratio for each GO term. The circle size reflects the number of genes. (**B**) GSEA-based GO enrichment plots by R package enrichplot. Green line indicates the running enrichment scores. The bar color represents the activated (red) or suppressed (blue) functions. (**C**) The connection of genes among the prominent GO functions by enrichGO function. The circle color represents the different regulations of genes with log FC values (red; upregulation, green; downregulation). The arrow indicates *CLDN11* gene.

**Figure 5 cells-11-02476-f005:**
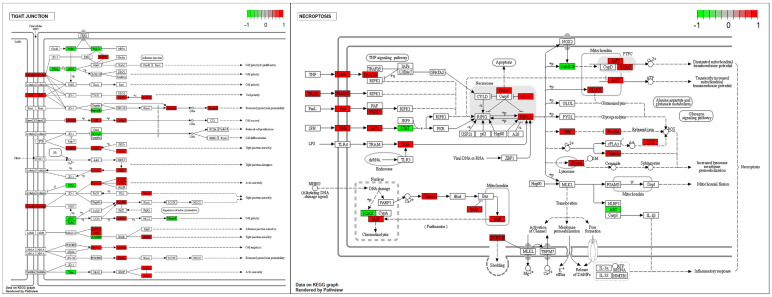
The expression pattern of DEGs involved in the tight junction and the necroptosis pathways. The DEGs were mapped by R package pathview. The box color represents the different regulations of genes with log FC value (red; upregulation, green; downregulation).

**Figure 6 cells-11-02476-f006:**
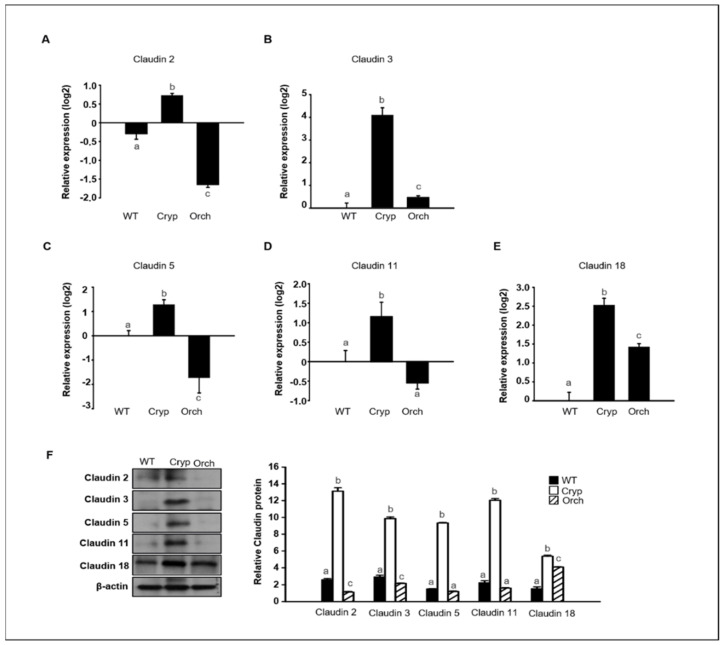
The relative expression of claudins in wild-type, cryptorchid, and orchiopexied dog testes. The expression of (**A**) *claudin-2*, (**B**) *claudin-3*, (**C**) *claudin-5*, (**D**) *claudin-11*, and (**E**) *claudin-18* genes were detected in wild-type, cryptorchid, and orchiopexied dog testes. The expression levels of these genes are significantly increased in cryptorchid testes compare to wild-type. Data are presented as means ± standard deviation. Values with different letters are significantly different (**F**) protein levels of claudins 2, 3, 5, 11, and 18 were also analyzed via immunoblot analysis. The values of all protein levels were normalized to that of β-actin. Results are expressed as mean ± standard deviation. Values with different letters are significantly different. These results are consistent with the data from our qPCR analysis.

**Figure 7 cells-11-02476-f007:**
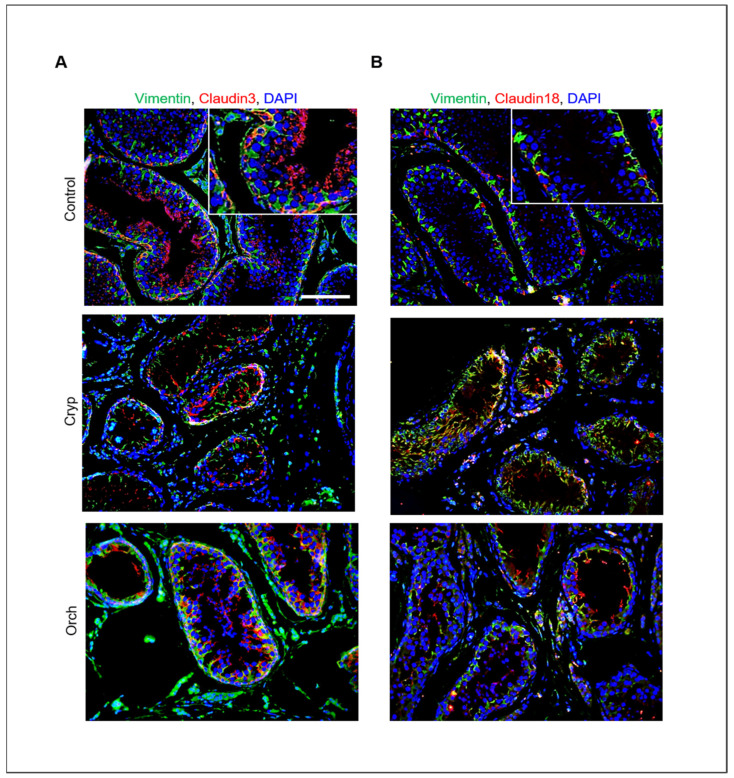
Relative expression of claudins 3 and 18 in wild-type, cryptorchid, and orchiopexied dog testes. (**A**) Vimentin and claudin-3. (**B**) Vimentin and claudin-18 were detected in wild-type, cryptorchid, and orchiopexied dog testes by immunostaining analysis. Vimentin, claudin-3, and claudin-18 were localized in the cell membrane near the basement membrane of the testes. Scale bars = 50 μm; *n* = 3, two pairs of testes.

**Figure 8 cells-11-02476-f008:**
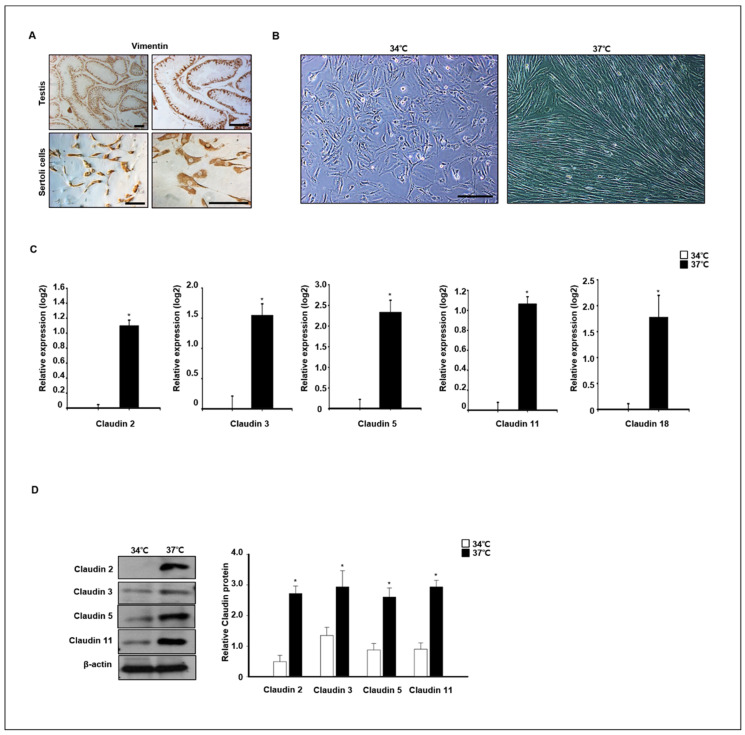
Effect of temperature on claudin expression in in vitro-cultured canine primary Sertoli cells. (**A**) Primary canine Sertoli cells were isolated from 12-month old dog testes. Isolated cells and whole testes were stained with vimentin antibodies as a Sertoli cell marker. (**B**) Images of the morphological change in primary Sertoli cells were taken from cells cultured at 34 and 37 °C. Scale bars = 50 μm. (**C**) The expression of claudins 2, 3, 5, 11, 18 was determined by qPCR from Sertoli cells cultured at 34 and 37 °C. These genes were detected at higher levels in cells cultured at 37 °C compared to the others. Results are expressed as mean ± standard deviation. * *p* < 0.05. (**D**) Protein levels of claudins 2, 3, 5, 11, and 18 were detected by immunoblot analysis. Consistently, the levels of these proteins were relatively increased in cells cultured at 37 °C compared to others. Quantitative data are expressed as mean ± standard deviation. * *p* < 0.05.

**Table 1 cells-11-02476-t001:** List of antibodies used in immunostaining.

Antibody	Company	Catalogue Number	Diluted
*Claudin 2*	Thermo Fisher	32-5600	1:200
*Claudin 3*	Thermo Fisher	34-1700	1:200
*Claudin 5*	Thermo Fisher	35-2500	1:200
*Claudin 11*	Thermo Fisher	36-4500	1:200
*Claudin 18*	Thermo Fisher	700178	1:200
*SYCP3*	abcam	Ab97672	1:200
*Sox9*	abcam	ab185966	1:200
*Vasa*	abcam	ab13840	1:200
*Vimentine*	Santa Cruz Biotech	SC-373717	1:200
*β-* *Actin*	Santa Cruz Biotech	SC-47778	1:200

**Table 2 cells-11-02476-t002:** Primers used for real-time reverse transcription–polymerase chain reaction of cDNA from canine samples.

Gene	Forward Primer	Reverse Primer
*Claudin 2*	5′- TGGGCATTATTTCCTCCTTG -3′	5′- AAACTCGCTCTTGGCTTTGG -3′
*Claudin 3*	5′- CTCATCGTCGTGTCCATCC-3′	5′- ATGGTGATCTTGGCCTTGG -3
*Claudin 5*	5′- CCATGTCGCAGAAGTACGAG -3	5′- ACTTGACCGGGAAGCTGAAG-3
*Claudin 11*	5′- GGGCTGTACCACTGCAAGC-3′	5′- CGTCAGCAGCAGGAGAATG-3′
*Claudin 18*	5′- GCCATCGGCCTCCTAGTATC -3′	5′- CAGAGGTCAGTGTCATGTTGG -3′
*SYCP3*	5′- CCCTCTGGAAGAAAGCACAC -3′	5′- CATCCTCCTCGGAACCTCTC -3′
*Stra8*	5′- GGAGTTGGAACAAACCTTGG -3′	5′- CGCCTTGACTTCCTCTAAGC -3′
*Vasa*	5′- TGGACGTACTGGTCGTTGTG -3′	5′- TCTTCCAACCATGCAGGAAC -3′
*Sox9*	5′- ACCACCCGGATTACAAGTACC -3′	5′- GGAAATGTGCGTCTGTTCG -3′
*WT1*	5′- CGAAAGTTCTCCCGGTCTG -3′	5′- GGTGCATGTTGTGATGACG -3′
*AMH*	5′- CCTGGAGGAAGTGACATGG -3′	5′- CAGGGTAGAGCACCAGCAG -3′
*GAPDH*	5′- AATTCCACGGCACAGTCAAG -3′	5′- TACTCAGCACCAGCATCACC -3′

## Data Availability

All data are contained within this article or Appendix A.

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
