# Peer review of "Transcriptomic Analysis of Testicular Gene Expression in a Dog Model of Experimentally Induced Cryptorchidism"

_cells, 2022, doi:10.3390/cells11162476_

Round 1

Reviewer 1 Report

In the Discussion section authors must clearly indicate the physiological relevance and applications of their results, as well as how claudins could be used as potential markers of male infertility. 

Author Response

Reviewer 1)

Comments and Suggestions for Authors

In the Discussion section authors must clearly indicate the physiological relevance and applications of their results, as well as how claudins could be used as potential markers of male infertility. 

Response: Thank you for reviewer`s comment. we additionally described in discussion section in response to reviewer’s comment (with blue color)

Reviewer 2 Report

The manuscript can be published in the present form.

Author Response

Reviewer 2)

Comments and Suggestions for Authors

The manuscript can be published in the present form.

Response: Thank you for reviewer`s comment 

This manuscript is a resubmission of an earlier submission. The following is a list of the peer review reports and author responses from that submission.

Round 1

Reviewer 1 Report

The revised manuscript is improved. There are only 3 small things that need to be corrected before publication.

Minor points:

Lines 459-460 - As we mentioned before and the authors agreed, the model was not established in this work. So, please, change the conclusion statement “we established a surgically induced cryptorchidism model in dogs and performed a comparative study of cryptorchid, orchiopexied, and wild-type testes” to “we performed a comparative study of surgically induced cryptorchid, orchiopexied, and wild-type testes in dogs.”

 Lines 295 and 300 – correct the word cryptorchid misspelled on these lines as “crytochid” and “cryptochid”

 I would suggest a language revision on the whole new discussion paragraph (lines 433-445). For instance, change “was differ with” to ‘differed from’ (on line 439), and change “the distinct temperature” to ‘distinct temperatures’ (on line 441).

Reviewer 2 Report

In the manuscript entitled “Transcriptomic Analysis of Testicular Gene Expression in Normal and Cryptorchid Dogs” the authors aim to describe the effect of heat on transcriptomic expression of Sertoli cells claudins 2, 3, 5, 11, and 18 in a dog with induced cryptorchidism.  The project idea could be interesting considering the importance of these proteins at level of blood-testis barrier that regulates spermatogenesis. The manuscript however shows several important flaws in the major parts of the manuscript. The text in the introduction describes the importance of cryptorchidism in both rat and in mouse models while no findings relative to differences and characteristics of canine cryptorchidism (while the title is focused on canine cryptorchidism) are introduced. The world dog or canine is not reported between key words. The methods are described in a very confusing way. The reference for the surgically induced cryptorchidism in dogs (ref.18) standardized describes dogs that were six to seven months of age while in the method section of the manuscript the dogs used are at 12 months of age. This condition certainly can influence the morphology and the hormonal profile. Moreover, the authors do not describe the total number of animals used for the study and the number of different groups of dogs examined (wt, cryp, orch). Also, in the results n. 3 sample used for the analysis is a too reduced number for significant results. In this form, the manuscript is difficult to read.

Reviewer 3 Report

The authors conducted an interesting study focused on the expression pattern of claudins in surgically induced cryptorchid testes from dogs. The research is well conducted, and the results are clearly presented. However, the conclusions paragraph must be improved to highlight the physiological significance of the results obtained.

Please, note that surgically induced cryptorchidism in adult males does not correlate with spontaneous cryptorchidism which results in lack of spermatogenesis instead of germ cell loss. Therefore, surgically induced cryptorchidism could be an appropriate model to analyse the effect of testicular heating on sperm quality, but the results obtained do not correlate with spontaneous cryptorchidism. Therefore, authors must review the whole text taking this in consideration to avoid misunderstandings.

On the other hand, authors must address some minor changes before the acceptance of the manuscript for publication.

Line 34. Please, note that the basal compartment only has spermatogonia. Once formed, primary spermatocytes migrate through the blood-testis barrier towards the apical compartment where they divide by meiosis I. Therefore, authors must review this sentence and provide an accurate message.

Subsection 2.1. Authors could include a scheme of the experimental design to improve the comprehension of this section.

Lines 220-221. “few testicular cells”. This is a very ambiguous term. Do the authors refer to spermatogonia? If so, please revise this sentence and use appropriate terminology.

Line 237. Please, correct “many cells” by “many germ cells”.

Line 462-463. Please, revise the content of this lines and improve the message to highlight the physiological relevance of the results obtained.